# Surface Crack Monitoring by Rayleigh Waves with a Piezoelectric-Polymer-Film Ultrasonic Transducer Array

**DOI:** 10.3390/s23052665

**Published:** 2023-02-28

**Authors:** Xiaotian Li, Voon-Kean Wong, Yasmin Mohamed Yousry, David Boon Kiang Lim, Percis Teena Christopher Subhodayam, Kui Yao, Liuyang Feng, Xudong Qian, Zheng Fan

**Affiliations:** 1Institute of Materials Research and Engineering, A*STAR (Agency for Science, Technology and Research), 2 Fusionopolis Way, Innovis #08-03, Singapore 138634, Singapore; 2Department of Civil and Environmental Engineering, Centre for Offshore Research and Engineering, National University of Singapore, Singapore 117576, Singapore; 3School of Mechanical and Aerospace Engineering, Nanyang Technological University, 50 Nanyang Avenue, Singapore 639798, Singapore

**Keywords:** ultrasonic, piezoelectric transducer array, welded joint, surface fatigue crack, structural health monitoring, reflected Rayleigh wave, PVDF

## Abstract

This paper presents a method for measuring surface cracks based on the analysis of Rayleigh waves in the frequency domain. The Rayleigh waves were detected by a Rayleigh wave receiver array made of a piezoelectric polyvinylidene fluoride (PVDF) film and enhanced by a delay-and-sum algorithm. This method employs the determined reflection factors of Rayleigh waves scattered at a surface fatigue crack to calculate the crack depth. In the frequency domain, the inverse scattering problem is solved by comparing the reflection factor of the Rayleigh waves between the measured and the theoretical curves. The experimental measurement results quantitatively matched the simulated surface crack depths. The advantages of using the low-profile Rayleigh wave receiver array made of a PVDF film for detecting the incident and reflected Rayleigh waves were analyzed in contrast with those of a Rayleigh wave receiver using a laser vibrometer and a conventional lead zirconate titanate (PZT) array. It was found that the Rayleigh waves propagating across the Rayleigh wave receiver array made of the PVDF film had a lower attenuation rate of 0.15 dB/mm compared to that of 0.30 dB/mm of the PZT array. Multiple Rayleigh wave receiver arrays made of the PVDF film were applied for monitoring surface fatigue crack initiation and propagation at welded joints under cyclic mechanical loading. Cracks with a depth range of 0.36–0.94 mm were successfully monitored.

## 1. Introduction

The fatigue crack in metals is the most common cause of failure in industry and has caused many catastrophic disasters. The fatigue crack usually develops from the surface of a structure, where fatigue damage typically initiates as shear cracks on crystallographic slip planes [1,2,3,4]. The surface fatigue crack growth proceeds in two steps: crack initiation and crack propagation, both of which may not be noticed macroscopically. In offshore and marine structures, the fatigue of welded joints is a common problem [5,6]. Surface fatigue cracks appear in welded structures mostly at the welded joints rather than in the base metal [7]. This is because the welding process includes heating and subsequent cooling as well as a fusion process with additional filler material, resulting in different materials and inherent metallurgical geometrical defects. Therefore, the fatigue behavior of welded joints has attracted great attention, and surface fatigue crack growth monitoring is considered one of the most effective ways to study the fatigue behavior and monitor the structural health of a welded structure [8].

To meet the safety requirement for structures, the detection of surface fatigue cracks before they reach a critical length is necessary. Ultrasonic testing is now widely used for its high sensitivity, high penetrating capability, high accuracy, and fast response [9]. Rayleigh waves, which are surface acoustic waves, are effective for surface crack detection. This is because when Rayleigh waves propagate along a surface, their amplitude decreases exponentially with the depth of surface irregularities, making them more sensitive to surface defects compared to other wave types. In this case, a crack depth and location can be estimated simultaneously by monitoring the Rayleigh waves scattered from the crack. The detailed characteristics of the crack can be obtained from the traveling time and amplitudes of the incident, reflected, and transmitted Rayleigh waves [10].

Because Rayleigh waves are sensitive to surface discontinuities, non-contact methods to detect the scattered Rayleigh waves are preferred, such as a laser interferometer and the EMAT (Electro Magnetic Acoustic Transducer) [11,12,13,14]. However, it is difficult to place a bulky equipment at welded joints with a complex structure. To detect surface fatigue cracks on welded joints, ultrasonic transducers with simplified structure, high accuracy, and low profile on the surface conditions are required.

In this study, a Rayleigh wave receiver array made of a thin film polyvinylidene fluoride (PVDF) ultrasonic transducer array comprising nine PVDF ultrasonic transducers is proposed to meet the above stated requirements for monitoring surface fatigue crack growth at welded joints. The depth of the surface fatigue cracks was estimated from Rayleigh waves generated and detected by the ultrasonic transducers, based on a reference reflection factor curve and an improved measurement procedure. The reference curve was established from analytical results, and a delay-and-sum algorithm was deployed in the measurement procedure to average out the unwanted modes. The proposed Rayleigh wave receiver array is made of a flexible and thin PVDF film bonded on the structure. Compared to the traditional ultrasonic transducer made of lead zirconate titanate (PZT), the PVDF film is lightweight, conformal to the surface of the structure, and low-profile, allowing the Rayleigh waves to propagate over the surface with low attenuation and the array receiver to detect the Rayleigh waves. This work presents novel methods for monitoring the depth of surface defects on a welded structure with a Rayleigh wave receiver array made of the PVDF film and a delay-and-sum algorithm, enabled by the low-attenuation Rayleigh wave receiver array design allowing the significant enhancement of incident and reflected Rayleigh waves.

The monitoring procedure was demonstrated experimentally with actual surface fatigue cracks. The incident and reflected Rayleigh waves propagating near the crack were separated from the other wave types by an enhanced delay-and-sum algorithm. The experimental results were similar to the simulation results when the crack depth ranged from 0.36 mm to 0.94 mm.

## 2. Crack depth Determination

### 2.1. Scattering of Rayleigh Waves by a Surface-Breaking Crack

The interaction between Rayleigh waves and cracks has been widely studied since the 1970s [15]. When Rayleigh waves are scattered at a crack, information in the scattered Rayleigh wave including arrival time, signal amplitude, and signal frequency can be used to estimate the crack depth. For example, the crack depth can be estimated from the propagating time of the Rayleigh wave along the cracks. This method, however, is effective only when the crack depth is greater than 0.8 times the Rayleigh wavelength, else there will be a time delay fluctuation. For shallow cracks, the relative amplitudes of the incident, reflected, and transmitted Rayleigh waves are more useful indicators.

Reflection and transmission coefficients have been studied with the numerical simulations of theoretical models. The reflection and transmission coefficients are obtained by comparing the Rayleigh wave reflected from a crack and the Rayleigh wave transmitted to the other side of the crack. Using the elastodynamic ray theory, Achenbach et al. derived exact and approximate solutions for high-frequency waves scattered at surface-breaking cracks and cracks near a free surface [16]. The incident surface wave scattered at a surface-breaking crack in two-dimensional geometry was also studied by Mendelsohn et al. [17]. With the development of the Finite Element Method (FEM), the scattering behavior can be investigated more accurately in more detail. Based on the theory developed by Mendelsohn, Masserey et al. successfully measured the surface crack depth on thick steel plates with good accuracy and repeatability, down to the smallest available ratio of crack depth to Rayleigh wavelength a/λ=0.15 [18]. For welded joints, as in our study, it is not practical to use the transmission coefficient for crack depth measurement. This is because of the required complex structure that prevents its installation on the other side of the welding line for welded joint surface crack monitoring. Therefore, the reflection coefficient is used in this work. The reflection coefficient is defined as the ratio of the reflected Rayleigh wave signal to the incident Rayleigh wave signal. All parameters and abbreviations used in this paper are listed in Table 1.

A delay-and-sum method is useful to increase the signal-to-noise ratio of a Rayleigh wave by averaging out the unwanted wave modes [19]. The process is conducted by:(i).Selecting the first Rayleigh wave receiver as a reference point(ii).Calculating the time delay caused by the additional propagation time for the Rayleigh wave to travel to the subsequent Rayleigh wave receivers from the reference point(iii).Adding the time delay to the time domain signals collected from the Rayleigh wave receivers so that the Rayleigh waves are in phase(iv).Summing the delayed time domain signals(v).Using the enhanced Rayleigh waves for the determination of the crack depth.

In this work, with the delay-and-sum algorithm, a parameter representing reflection, called reflection factor (*F_r_*) was used and corresponds to
(1)$$F_{r}=\frac{\sum_{n}^{N}\left[\mathrm{FFT}(R_{r,n})/\max\left(\mathrm{FFT}\left(R_{r,n}\right)\right)\right]}{\sum_{n}^{N}\left[\mathrm{FFT}(R_{i,n})/\max\left(\mathrm{FFT}\left(R_{i,n}\right)\right)\right]},$$
where $R_{i,n}$ is the gated ultrasonic signal of the incident Rayleigh wave received by the n-*th* Rayleigh wave receiver, $R_{r,n}$ is the gated ultrasonic signal of the Rayleigh wave reflected from the crack and then received by the n-*th* Rayleigh wave receiver, and N is the total number of Rayleigh wave receivers. The gated ultrasonic signals were obtained by isolating the Rayleigh waves from unwanted ultrasonic signals in the time domain, collected by the Rayleigh wave receivers. The gated ultrasonic signals were then converted to the frequency domain using the Fast Fourier Transform (FFT). Ultrasonic signals attenuate the further they propagate. In order to prevent the unwanted wave types detected by the first Rayleigh wave receiver from overshadowing the Rayleigh wave detected by the subsequent Rayleigh wave receivers, each frequency spectrum of the gated ultrasonic signal was normalized before summation. As a result, the reflection factor $F_{r}$ used in this study could be larger than 1. Then, 2-D finite element models were applied to obtain the reflection coefficient for a notch with a certain depth at different central frequencies. The results were plotted with respect to the ratio of the crack depth to the wavelengt, and were in agreement with the prediction from Masserey et al. [17]. Note that this reflection coefficient was simulated without normalization, thus Cr < 1.

### 2.2. Characterization in the Frequency Domain

According to Figure 1, the reflection coefficient Cr reaches the maximum when a/λ=0.45. By measuring the maximum Cr and the corresponding frequency, we can obtain the wavelength and then calculate the crack depth. We have
(2)$$a=0.45\lambda =\frac{0.45v}{f},$$
where a is the crack depth, λ is the wavelength, v is the speed of sound in the medium, and f is the frequency corresponding to the maximum Cr. Since the Rayleigh wave velocity here is a constant, the surface defect depth can be rewritten as a function of the frequency. The relationship between the surface defect depth and the optimal frequency is shown in Figure 2. The relationship between the surface defect depth and the frequency that corresponds to the maximum Cr is shown in Figure 2.

## 3. Experimental Details

### 3.1. Ultrasonic Transducers’ Design

A schematic of the active ultrasonic transducer design for crack depth measurement using Rayleigh waves is shown in Figure 3.

For this design, the Rayleigh waves were generated with a Rayleigh wave transmitter. The Rayleigh wave transmitter consisted of an ultrasonic transducer installed on an ultrasonic wedge with a critical angle for Rayleigh wave generation, and the ultrasonic wedge was bonded on the surface of the structure with adhesive. Meanwhile, a Rayleigh wave receiver array made of a PVDF film with three discrete electrodes was bonded on the surface of the structure in front of the crack. A longitudinal wave emitted from the ultrasonic transmitter was converted into a Rayleigh wave at the interface of the structure and the wedge. As the Rayleigh wave traveled past the Rayleigh wave receiver array, an incident Rayleigh wave was detected by the Rayleigh wave receiver array. As the Rayleigh wave traveled farther, it was scattered at the crack when it reached the crack. Thereafter, the scattered wave was received by the Rayleigh wave receiver array. Information about the crack was contained in the scattered wave. Subsequently, the delay-and-sum algorithm of the detected received ultrasonic waves was applied. By repeating the process and analyzing the incident and reflected Rayleigh waves in the frequency domain, the depth and location of the crack could be monitored.

### 3.2. Structure to Be Monitored

In our experiments, six sets Rayleigh wave receiver arrays (labeled as ‘A’, ‘B’, ‘C’, ‘D’, ‘E’, and ‘F’) were bonded on the two surfaces of the structure to be monitored. Each of the Rayleigh wave receiver array contained three electrodes (named ‘1′, ‘2′, and ‘3′) that enabled the detection of Rayleigh waves in three different positions. The Rayleigh waves collected at each of the three positions were processed according to Equation (1) for signal enhancement. The schematic drawing of the transducers and a photo of the experimental setup with a welded joint to be monitored are presented in Figure 4.

The structure used in this study consisted of 38 mm thick cruciform welded S550 high-strength steel joints with good welding properties [20]. The structure was subjected to three-point bending on the top of the attachment plate. The maximum load applied was 50 kN. with a load ratio of 0.1, and the loading frequency was 3 Hz. As shown in Figure 4b, potential cracks occurred in the two bottom areas in the plate near the weld toe. Two Rayleigh wave transmitters were placed on two sides of the structure, as shown in Figure 4. Three sets of the Rayleigh wave receiver arrays made of the PVDF film were placed in front of each of the two Rayleigh wave transmitters.

As shown in Figure 4a, each Rayleigh wave receiver array comprised a PVDF film with three discrete linear electrodes that were equally spaced. The width of the electrodes was smaller than half of the Rayleigh wavelength, which was determined by the smallest crack depth to be monitored. If the width of the electrodes were larger than the half wavelength of the Rayleigh wave travelling across the width direction of the electrode, the Rayleigh wave signal collected by each electrode would destructively cancel out. To avoid this, the width of the transducers was no more than half of the Rayleigh wavelength.

### 3.3. Materials and Methods

An ultrasonic transducer (V106, Olympus Scientific Solutions Americas Inc. Waltham, MA, USA) with a central frequency of 2.25 MHz was used as the ultrasonic transducer for the Rayleigh wave transmitter. The ultrasonic transducer was bonded to the wedge using a conductive silver epoxy adhesive (Polytec PT EC244, Waldbronn, Germany), followed by curing at 70 °C for 1 h. The wedge was then bonded to the structure to be monitored by an epoxy adhesive layer (Araldite 2011) and cured overnight at room temperature.

For the Rayleigh wave receiver array, a PVDF film (PolyK, Philipsburg, PA, USA) with a thickness of 50 μm was bonded to the structure to be monitored using the conductive silver epoxy adhesive, followed by curing at 70 °C for 1 h. Afterward, a sticker mask was used to pattern nine discrete top electrodes on the bonded PVDF film. Silver electrodes were deposited by spraying, followed by curing at 70 °C for 10 min. The cured silver electrodes had a thickness of 5 μm. A waterproof layer was coated on top of the ultrasonic sensors as a protective layer. The electrodes were connected to a customized flexible printed circuit (MFS Technology, Singapore) and terminated to a D-Subminiature (DB9) connector, which was then connected to the ultrasonic testing system (Vantage 64, Verasonics Inc., Kirkland, WA, USA).

The Rayleigh wave receiver array made of the PVDF film with three discrete electrodes was compared with a laser interferometer (UHF 120, Polytec, Waldbronn, Germany) and a Rayleigh wave receiver array made of three discrete PZTs. The experimental setups of the Rayleigh wave receivers made of the piezoelectric polymer film ultrasonic transducer, the piezoelectric ceramic ultrasonic transducer, and the laser interferometer are shown in Figure 5. An excitation signal using a −13 Vp pulse with a pulse width of 385 ns was applied to the Rayleigh wave transmitter. A surface defect was simulated by machining a slot with 1 mm depth on a 40 mm thick aluminum block.

## 4. Results and Discussion

The generated Rayleigh wave would propagate along the surface, reach the Rayleigh wave receiver and be detected by the Rayleigh wave receiver showing the incident Rayleigh wave. The Rayleigh wave then would continue propagating to the crack, be reflected by the crack, and be received by the Rayleigh wave receiver array as a reflected Rayleigh wave. The incident and reflected Rayleigh waves detected by the Rayleigh wave receiver made of the PVDF film on the machined slot with a depth of 1 mm are presented in Figure 6. The received ultrasonic signals were analyzed using the delay-and-sum algorithm. The time window gating was set to be narrow to exclude noise and unwanted wave types. The gated ultrasonic signal was then processed with zero padding to ensure sufficient FFT points. Throughout the whole experiment, the total time period of the gated ultrasonic signal was set to 3.5 μs for both incident and reflected Rayleigh waves.

The performance of the Rayleigh wave detection system with the Rayleigh receiver array made of the PVDF film was compared with those of the Rayleigh wave receiver array made of PZT and the laser interferometer. Figure 7 presents the frequency spectra of the incident and reflected Rayleigh wave signals after applying the delay-and-sum algorithm for the three types of Rayleigh wave receivers. The reflection factor ($F_{r}$) was calculated by applying Equation (1), shown in Figure 7. For the defect depth determination, the $F_{r}$ peak was searched within the −6 dB frequency bandwidth of the incident Rayleigh wave.

No peaks were detected for the $F_{r}$ using the Rayleigh wave receiver array made of three discrete PZTs within the effective frequency bandwidth. This was due to the low reflected Rayleigh wave signal amplitude, which resulted from the high attenuation rate when the Rayleigh wave propagated past each PZT. For comparison, the amplitude of the incident Rayleigh waves detected by different types of Rayleigh wave receivers was recorded. The performance comparison for the three Rayleigh wave receiver array designs is presented in Table 2.

In Table 2, the Rayleigh wave receiver array made of the three discrete PZTs had the highest attenuation rate, while the Rayleigh wave receiver array made of the PVDF film with three discrete electrodes had an attenuation rate just slightly higher than that of the laser interferometer. The reason for the higher attenuation rate when using the Rayleigh wave receiver array made of discrete PZTs was the attenuation of the surface-sensitive Rayleigh wave when travelling across the bulky PZTs. Furthermore, due to the low mechanical damping properties of PZT, the frequency bandwidth of the detected Rayleigh waves was narrower. This led to a narrower surface defect depth measurement range. Therefore, the Rayleigh wave receiver array made of three discrete PZT strips bonded on the structure was not able to measure the depth of the defect. On the other hand, the feasibility of applying the PVDF film as a Rayleigh wave receiver for determining the depth of surface crack was successfully demonstrated via the experimental results. It was also demonstrated that the Rayleigh wave receiver array made of the PVDF film had a comparable incident Rayleigh wave frequency bandwidth as the non-contact laser interferometer. Despite showing a relatively lower crack depth measurement accuracy when compared to the laser interferometer, the Rayleigh wave receiver array made of a piezoelectric polymer layer has a much lower cost and does not require a line of sight to perform structural health monitoring.

The Rayleigh wave receiver array made of the PVDF film that was fabricated using the method described in the Section 3.3 was subsequently applied for surface fatigue crack monitoring, as shown in Figure 4 and as described before. The crack depth measurement range was determined from the −6 dB cutoff frequencies of the incident Rayleigh wave frequency spectra according to the delay-and-sum algorithm for the respective positions (Figure 8). After the lower and the upper cutoff frequencies were determined, the crack depth measurement range for each position was calculated using Equation (2), as summarized in Table 3.

For every 5000 fatigue loading cycles, ultrasonic signals were generated and received using the ultrasonic testing system. The collected ultrasonic signals were processed by applying the delay-and-sum algorithm to the incident and reflected Rayleigh waves. Thereafter, the peak of the $F_{r}$ was recorded, so that crack depth could be calculated. Figure 9 shows the Rayleigh wave spectral method with the delay-and-sum algorithm for position B at cycle 25,669. The incident and reflected Rayleigh wave signals detected by the Rayleigh wave receiver array in position ‘B’ are shown in Figure 9a,b. The signals were then processed using the delay-and-sum algorithm according to Equation (1), and the calculated $F_{r}$ is shown in Figure 9c. Thus, the optimal frequency was 2.33 MHz, and the crack depth was calculated as 0.595 mm using Equation (2).

The crack depth measured at each position for different fatigue loading cycles is presented in Figure 10. Figure 10a shows the crack depth monitored by the Rayleigh wave receiver array in positions ‘A’, ‘B’, and ‘C’; Figure 10b shows the crack depth monitored by the Rayleigh wave receiver array in positions ‘D’, ‘E’, and ‘F’. As shown in Table 2, the measurement range of each receiver array was limited by the −6 dB frequency bandwidth of the incident Rayleigh wave. For the Rayleigh wave receiver array ‘A’, the measurement range was 0.36 mm–0.94 mm, and the actual crack depth measured was 0.42 mm–0.92 mm. The crack depths measured from the six Rayleigh wave receiver arrays were different. This indicated that the cracks had a different growth behavior at different positions. With the six receiver arrays, crack information over this area was obtained. The crack depth was also simulated with the same method as in [21]. The theoretical analysis simulated crack growth of the weldline, i.e., positions ‘B’ and ‘E’. As the simulation was ideal, the simulation results of the two sides were symmetrical. Figure 10 shows that the simulation results had the same trend as the measured results of the crack depth. The numerical simulation, however, did not reflect the exact unsymmetrical cracks in the two sides of the welded joints, due to the simplified model of the weld geometry and boundary conditions. As a result, the simulation result tended to overestimate the crack growth due to the higher stress concentration in the finite element model in comparison to that in the experimental tests. Nonetheless, the results of the numerical simulation provided a reference for the crack measurement by the Rayleigh wave receiver arrays.

In recent years, piezoelectric ultrasonic transducers can be fabricated in situ on the surface of a structure to conduct structural health monitoring (SHM). These low-profile, lightweight and highly conformable piezoelectric ultrasonic transducers are known as direct-write ultrasonic transducers and have been demonstrated to be efficient for SHM, such as for detecting the presence of various defects and plastic deformation [22,23]. By combining the scalable direct-write ultrasonic transducer technology, an active ultrasonic SHM system with edge computing capability, and the methods proposed in this work, it will be possible to realize a scalable quantitative crack size monitoring over a large area [24,25,26].

## 5. Conclusions

Surface crack depth measurement using piezoelectric polymer film ultrasonic transducer arrays was successfully demonstrated by analyzing the detected Rayleigh waves in the frequency domain. A delay-and-sum algorithm was adopted for enhancing the detected Rayleigh wave signals. The proposed method was compared with Rayleigh wave detection with a Rayleigh wave receiver made of piezoelectric ceramic and a laser interferometer. Both methods using laser without physical contact and the low-profile PVDF film transducer array with minimized interference with the surface conditions were able to determine the crack depth because of the corresponding lower attenuation of the surface-sensitive Rayleigh waves (0.15 dB/mm). Meanwhile, the method using the bulky piezoelectric ceramic Rayleigh wave receiver arrays failed due to the significant attenuation of the Rayleigh waves (0.30 dB/mm). This method based on the frequency spectra of reflected Rayleigh waves is very sensitive for measuring crack depths below 1 mm, which makes it suitable for monitoring crack initiation and early-stage propagation. Furthermore, this method does not require the transmission of the ultrasonic signal across the structure and is thus applicable to measure cracks present in complex structures such as welded joints. Compared to the laser interferometer, the Rayleigh wave receiver array made of a piezoelectric polymer layer is of much lower cost and does not require a line of sight to perform crack monitoring.

The crack depth monitoring procedure using six PVDF Rayleigh wave receiver arrays was further demonstrated by measuring real fatigue cracks induced at two welded joints of a 38 mm thick steel cruciform structure under cyclic mechanical loading test. The maximum load applied was 50 kN with a load ratio of 0.1, and the loading frequency was 3 Hz. Real-time monitoring was realized with the six PVDF Rayleigh wave receiver arrays bonded to the structure. The measurement range was determined by the −6dB frequency bandwidth of the incident Rayleigh wave. We showed that the Raleigh wave receiver arrays worked efficiently from 7140 to 36,700 fatigue loading cycles. Crack initiation and propagation along the two welded joints with a depth range between 0.36 and 0.94 mm were determined quantitatively during the cyclic fatigue loading process.

## Figures and Tables

**Figure 1 sensors-23-02665-f001:**
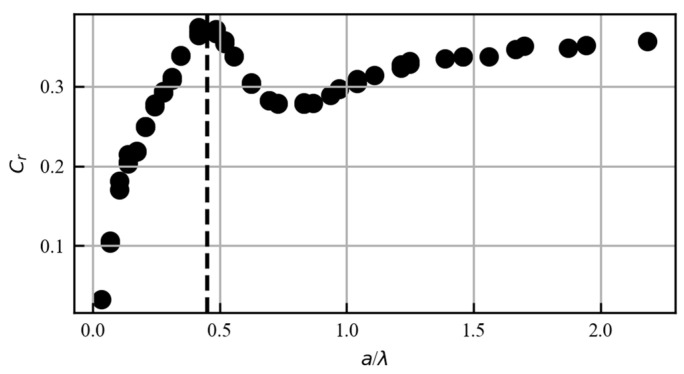
Reflection coefficient ($C_{r}$) by FEM as a function of different depth-to-wavelength ratios (a/λ). The vertical dashed line corresponds to a/λ=0.45.

**Figure 2 sensors-23-02665-f002:**
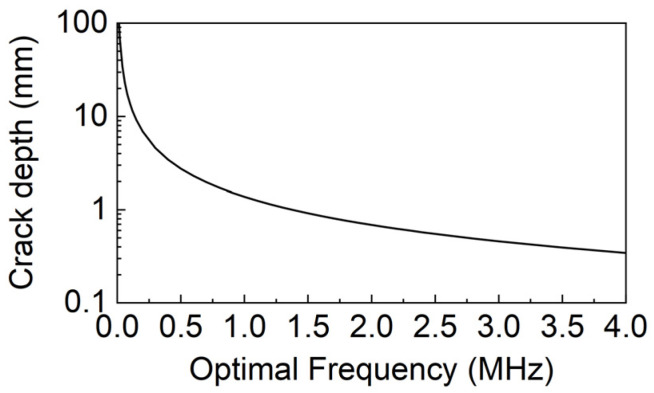
Relationship between the crack depth and the frequency corresponding to the maximum Cr.

**Figure 3 sensors-23-02665-f003:**
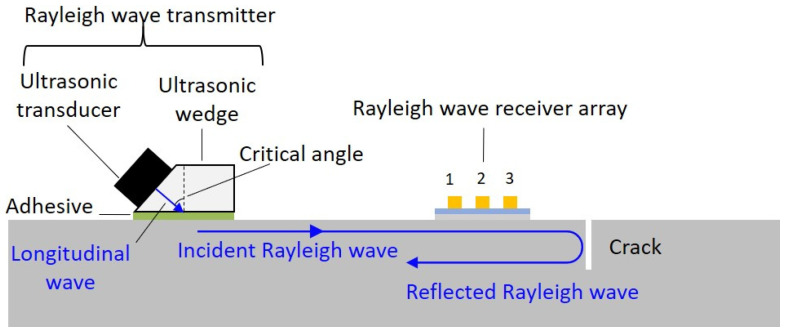
Schematic illustration of the system for monitoring the depth of a surface crack on a structure with Rayleigh waves, where the Rayleigh wave receiver array is positioned between the Rayleigh wave transmitter and the surface crack. The Rayleigh wave transmitter generates Rayleigh waves, and the Rayleigh wave receiver array detects both the incident Rayleigh wave and the Rayleigh wave reflected from the edge of the surface crack.

**Figure 4 sensors-23-02665-f004:**
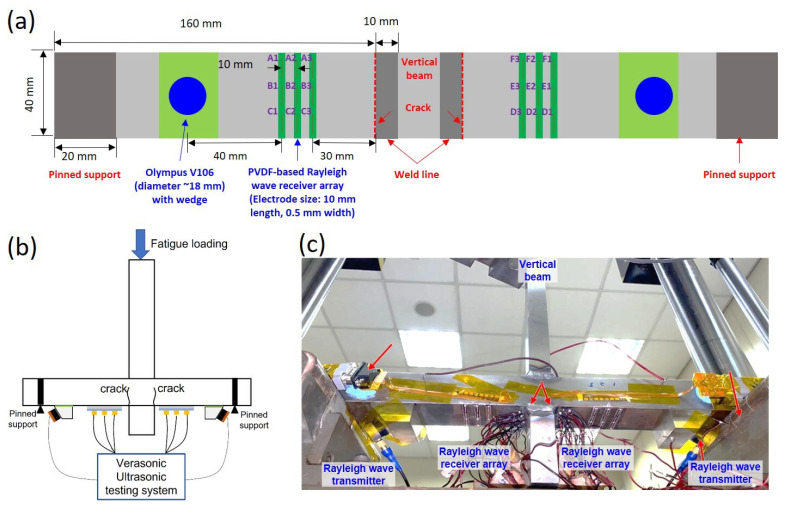
Test setup. (**a**) Schematics of the placement of the ultrasonic transducers and dimensions of the PVDF-based Rayleigh wave receiver arrays fabricated on a welded cruciform joint structure, viewed from the bottom of the fatigue loading test setup; (**b**) schematic drawing of the experimental setup and data collection; and (**c**) picture of the fatigue loading test setup in the lab.

**Figure 5 sensors-23-02665-f005:**
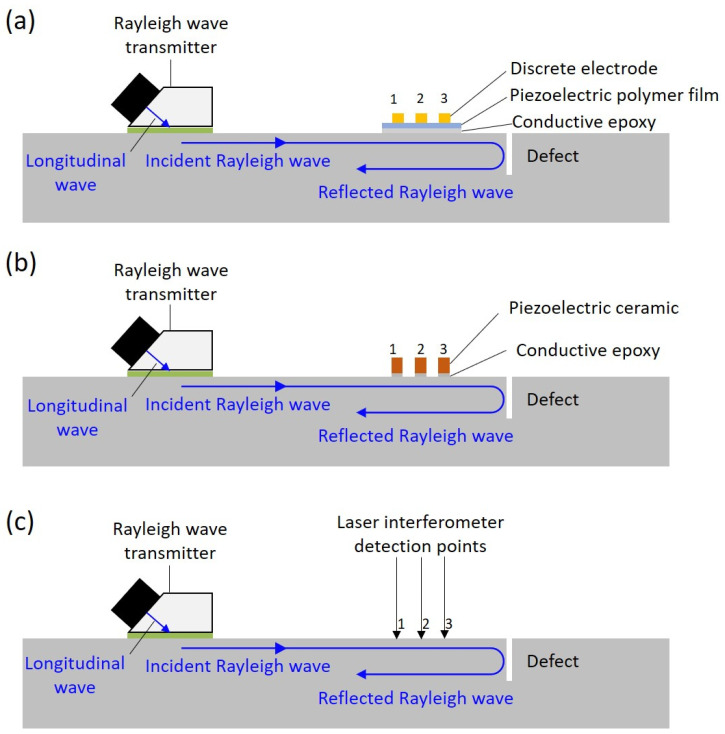
Experimental setups for monitoring the depth of a surface defect on a structure with Rayleigh ultrasonic waves, using an ultrasonic transducer with angle wedge as the Rayleigh wave transmitter; the Rayleigh wave was detected by (**a**) a Rayleigh wave receiver array made of a PVDF film and three discrete electrodes; (**b**) three discrete PZTs; and (**c**) a laser interferometer.

**Figure 6 sensors-23-02665-f006:**
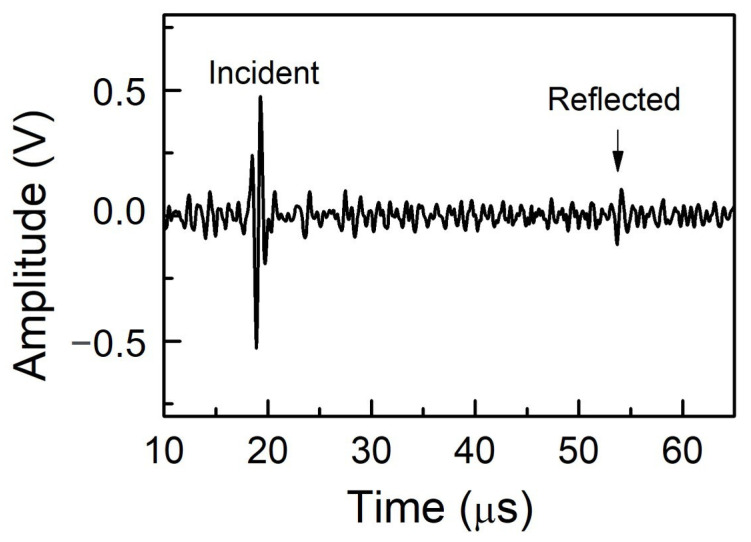
Incident and reflected Rayleigh waves detected by the Rayleigh wave receiver made of the PVDF film.

**Figure 7 sensors-23-02665-f007:**
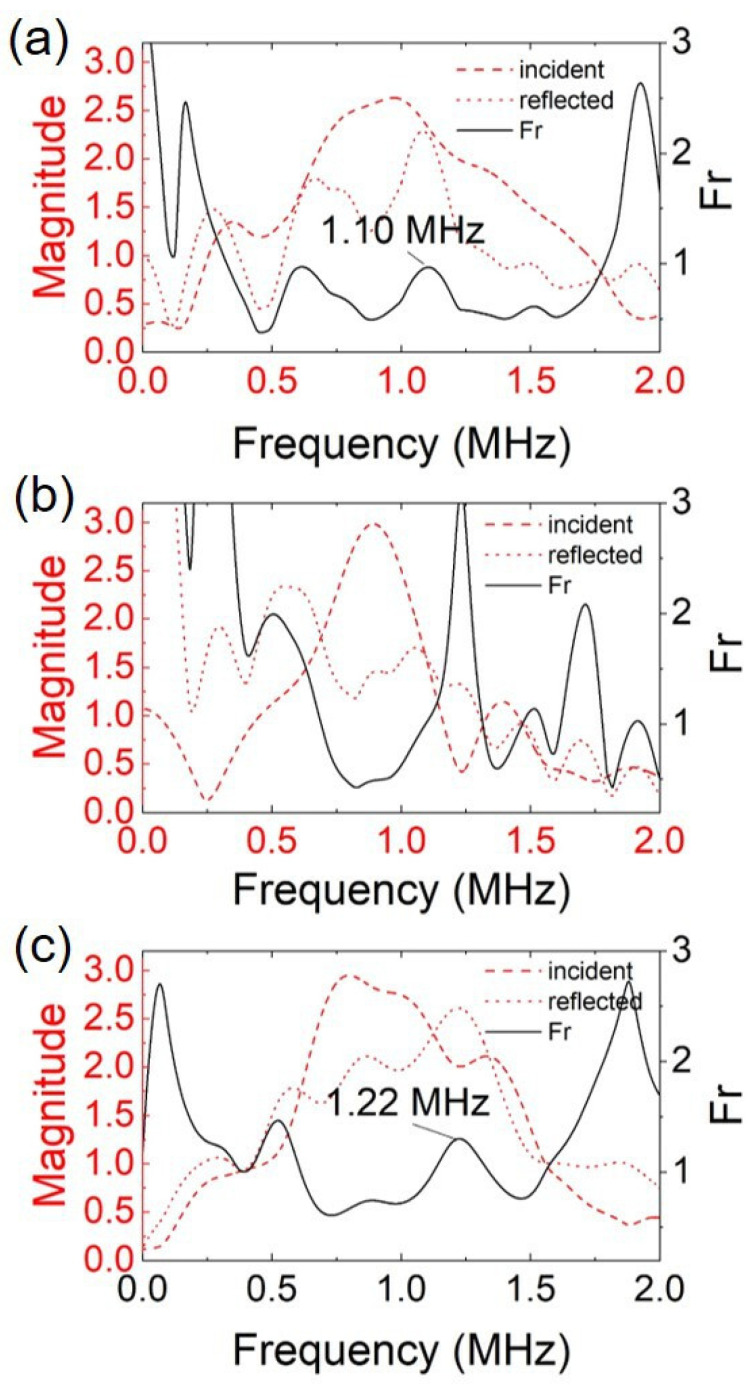
Frequency spectra of the incident Rayleigh wave and reflected Rayleigh wave and reflection factor ($F_{r}$) detected by Rayleigh wave receivers made of (**a**) the Rayleigh wave receiver array made of the PVDF film and three discrete electrodes; (**b**) the Rayleigh wave receiver array made of three discrete PZTs; and (**c**) the laser interferometer.

**Figure 8 sensors-23-02665-f008:**
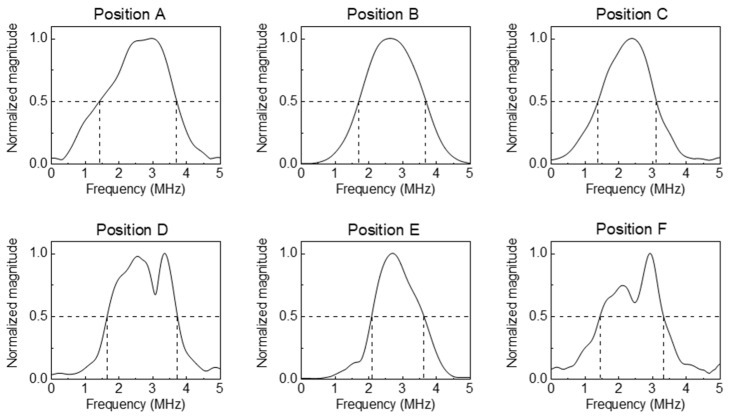
Frequency spectra of the incident Rayleigh wave after the delay-and-sum algorithm for the six positions as shown in Figure 4.

**Figure 9 sensors-23-02665-f009:**
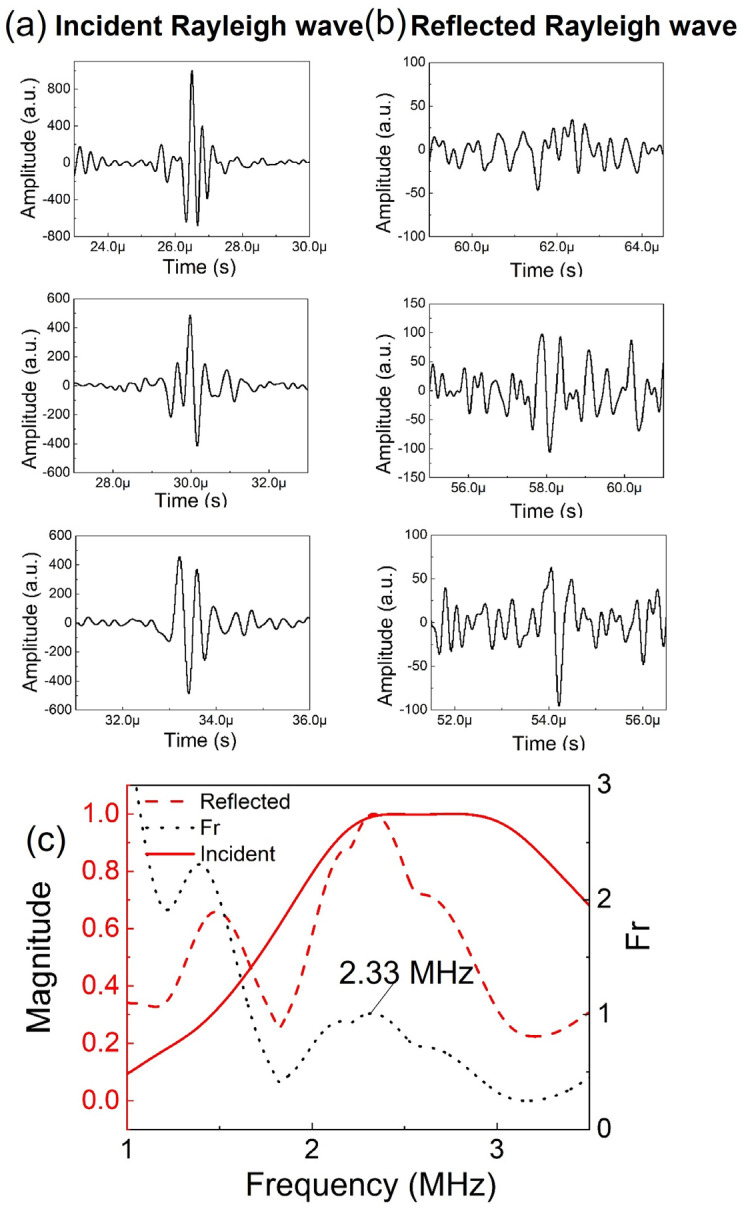
Signal processing for position B at cycle 25,669: (**a**) incident Rayleigh wave signals detected by B1, B2, and B3; (**b**) reflected Rayleigh wave signals detected by B1, B2, and B3; (**c**) frequency spectra for incident Rayleigh wave and reflected Rayleigh wave according to the delay-and-sum algorithm, and calculated $F_{r}$.

**Figure 10 sensors-23-02665-f010:**
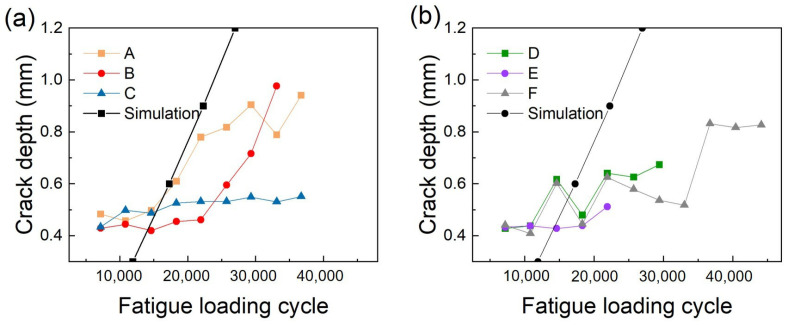
Crack depth monitoring results at positions (**a**) ‘A’, ‘B’, ‘C’ and simulation of ‘B’; (**b**) ‘D’, ‘E’, ‘F’ and simulation of ‘E’.

**Table 1 sensors-23-02665-t001:** List of symbols and notations used in this work.

a	Crack depth
λ	Rayleigh wavelength
$R_{i}$	Incident Rayleigh wave
$R_{r}$	Reflected Rayleigh wave
$F_{r}$	Reflection factor
N	Number of Rayleigh wave receivers
v	Rayleigh wave speed
$C_{r}$	Reflection coefficient
f	Frequency corresponding to the maximum $C_{r}$

**Table 2 sensors-23-02665-t002:** Performance benchmark for using the PVDF film, the discrete PZT, and the laser interferometer as Rayleigh wave receivers to measure 1 mm deep surface defects.

Rayleigh Wave Receiver	Incident Frequency Bandwidth (MHz)	Attenuation Rate (dB/mm)	Measured Defect Depth (mm)
PVDF film	0.83	0.15	1.21
Discrete PZT	0.47	0.30	Unable to measure
Laser interferometer	0.90	0.08	1.09

**Table 3 sensors-23-02665-t003:** Sensor positions and crack depth measurement ranges.

Position	Crack Depth Measurement Range (mm)
A	0.36–0.94
B	0.36–0.78
C	0.43–0.96
D	0.36–0.81
E	0.37–0.64
F	0.40–0.93

## Data Availability

Not applicable.

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
