# Peer review of "Surface Crack Monitoring by Rayleigh Waves with a Piezoelectric-Polymer-Film Ultrasonic Transducer Array"

_sensors, 2023, doi:10.3390/s23052665_

Round 1

Reviewer 1 Report

This authors compare three different techniques for for sizing surface cracks using Rayleigh waves. Namely, Rayleigh wave receiver array made of PVDF film, laser vibrometry and array of piezoelectric transducer. The method employs the determined reflection coefficient for Rayleigh waves scattered by a surface  breaking fatigue crack to estimate the crack depth.

The reviewer is not familiar with term "the gated ultrasonic signal". Is it a windowed signal?

Figure 3c is of bad quality and it is difficult to distinguish details of the experimental setup.

The description of the manufacturing and the design of the proposed Rayleigh wave receiver array is given in the manuscript, but an additional figure with schematic design/geometry of the proposed Rayleigh wave receiver array with all the details is strongly recommended.

Positions A-F are not clearly shown in Figure 4. Please clarify this point.

Reviewer 2 Report

The presented manuscript seems to be interesting for readers of the Sensors journal, it is written in a good manner and suits the requirements of the journal. It can be accepted for publication after minor corrections listed below. 

- The "Abstract" section should contain the main achievements of research not general discussion. Re-organization of abstract is needed.

- Abbreviation/ acronyms, should all be defined at their first occurrence in the manuscript

- All parameters used in formulas must be explained. It is recommended to attach all parameters and abbreviations used in a table at the end of the article.

- The novelty of work at the end of manuscript “introduction” is not sufficient and should be explained more.

- The caption of "Figure 81." should be modified to "Figure 8."

- The caption of Figure 10 should be completed. Regarding the difference between simulation and practical results, more explanations should be provided.

-The optimum condition in the fabrication operation needs to be determined. The authors need to pay attention in the revision stage.

- In the "Conclusion" section, the authors should present more quantitative data as the main results of the research study rather than just some qualitative data.

- Literature review is not sufficient and authors must review and cite more papers in the field of “Predicting and modeling the failure of graded materials and composites” and especially newly published ones. Doing this, review and citing the following refs could be helpful: Mathematical and Computer Modelling, 55(3-4), pp.1339-1353.

Round 2

Reviewer 1 Report

The authors improved the manuscript except for some minor imperfections:

- The quality of Figure 4 is still very poor to distinguish details.

- Thicknesses of sub-layers in the Rayleigh wave receiver array made of PVDF film are not given in the manuscript (perhaps the reviewer have missed it).

Author Response

The authors improved the manuscript except for some minor imperfections:

Point 1: The quality of Figure 4 is still very poor to distinguish details.

Response 1: The authors accepted the reviewer’s comments on Figure 4 that can be further improved. The improved Figure 4 is provided in the revised manuscript.

Point 2: Thicknesses of sub-layers in the Rayleigh wave receiver array made of PVDF film are not given in the manuscript (perhaps the reviewer have missed it).

Response: In response to the reviewer’ comments, detailed thicknesses are given in the revised manuscript in Lines 231-236:

“For the Rayleigh wave receiver array, PVDF film (PolyK, PA, USA) with a thickness of 50 μm is bonded to the structure to be monitored using the conductive silver epoxy, followed by curing at 70 °C for 1 hour. Afterward, a sticker mask is used to pattern nine discrete top electrodes on the bonded PVDF film. Silver electrodes are deposited by spraying process, followed by curing at 70 °C for 10 minutes. The cured silver electrodes have thicknesses of 5 μm. A waterproof layer is coated on top of the ultrasonic sensors as protective layer.”

Reviewer 2 Report

As authors have performed an adequate revise, the manuscript might be accepted for publication in the journal. 

Author Response

As authors have performed an adequate revise, the manuscript might be accepted for publication in the journal. 

We thank the reviewer for the positive comments.